# Neurologic manifestations of COVID-19 and viral test in cerebrospinal fluid

Carla de Oliveira Cardoso[ID][1]*, Evandra Strazza Rodrigues Sandoval[2¤],
Lilian Beatriz Moreira de Oliveira Chagas[2], Soraya Jabur Badra[1], Dimas Tadeu Covas[2],
Simone Kashima Haddad[2], Luiz Tadeu Moraes Figueiredo[1]

1 Virology Research Center, Department of Internal Medicine, Medical School, University of São Paulo, Ribeirão Preto, São Paulo, Brazil, 2 Advanced Molecular Biology Laboratory, Blood Center of Ribeirão Preto, Medical School, University of São Paulo, Ribeirão Preto, São Paulo, Brazil

¤ Current Address: Faculdade de Medicina de Ribeirão Preto (FMRP), Campus USP, Ribeirão Preto, São Paulo, Brazil
* carla_cardoso@hotmail.com (COC)

## Abstract

### Background

Neurological manifestations are present in about one-third of COVID-19 cases, ranging from mild symptoms, such as anosmia, to more severe forms like demyelinating syndromes. Although direct invasion of the CNS has been demonstrated, the immune- mediated pathway is also described and more accepted. Even in cases where viral detection in CSF is absent, it should not rule out neuroinvasion. There are few prospective studies about neurological manifestations of COVID-19, especially with viral tests in CSF; as well there are still many questions about COVID-19 associated with neurological disease. Thus, we describe clinical and CSF findings of a prospective cohort of patients with nasal positive tests for SARS-CoV-2 and neurological involvement. We also discuss the pathogenic mechanisms related to these manifestations.

### Methods and Findings

This is a prospective cohort study; 27 patients were evaluated according to clinical presentation, the time interval between COVID-19 diagnosis and onset of neurological alterations, syndromic diagnosis, imaging and CSF findings. Real time polymerase chain reaction for SARS-CoV-2 genome was performed in all CSF samples. 2 RT-PCR in spinal cord fluid resulted positive in 9 (33.3%) cases, five of them had a positive swab nasal test concomitant to neurologic disease. Respiratory signs were described in 12 out 27 patients, five of them with viral detection in CSF. White cell counts in CSF were normal range in the majority of cases, except for 3 occurrences: two patients had elevated CSF WBC counts and viral detection in CSF (10 and 36 cells/mm3) and one also had elevated CSF WBC count but viral detection in CSF was negative (21cells/mm3). The observed neurological signs encompassed a diverse neurologic spectrum, including seizures, paresis, gait abnormalities, headaches, alteration in consciousness and memory or cognitive impairment. Both imaging and CSF alterations exhibited non-specific characteristics. Syndromic

**Data availability statement:** All relevant data are within the manuscript.

**Funding:** The cohort study was supported by Fundação de Amparo à Pesquisa de São Paulo, which maintains almost all researches from Virology Research Center. The Foundation is not involved in the design of the project, the collection or analysis of data, and is not involved in the publication of the final work. Neither I nor the other authors have been paid by this foundation for their work The funders had no role in study design, data collection and analysis, decision to publish, or preparation of the manuscript.

**Competing interests:** The authors have declared that no competing interests exist.

diagnoses included stroke, dementia or cognitive impairments, Guillain-Barré Syndrome, encephalitis, encephalomyelitis, acute flaccid palsy and optical neuritis.

## Conclusions

The patients in the present study had COVID-19 and neurologic involvement including a wide range of clinical manifestations. SARS-CoV-2 was detected in one-third of CSF samples, regardless of time interval between COVID-19 diagnosis and the onset of neurological signs. These conditions encompass various pathogenic pathways and the neuroinvasion potential of SARS-CoV-2 should be more studied.

## Introduction

In December 2019, the first case of COVID-19 (coronavirus disease-19) emerged in Wuhan, China, and it quickly reached pandemic status. Although it is known as acute respiratory disease, the involvement of other organs and systems has been well-demonstrated [1–5]. Neurological manifestations are present in about one-third of COVID-19 cases, ranging from mild symptoms, such as anosmia, to more severe forms like encephalitis and demyelinating syndromes [6]

Neurological involvement of COVID-19 affects both central nervous system (CNS) and peripheral nervous system (PNS) [7]. They can be nonspecific, such as headache, anosmia, dysgeusia, seizures and consciousness alterations, or specific, like Guillain-Barré syndrome, transverse myelitis, encephalitis, CNS thrombosis, stroke, optical neuromyelitis [8–12]. These manifestations occur even during the typical respiratory phase of COVID-19 or within a few days to two months after the respiratory disease [13]. In some cases, neurological symptoms are the initial presentation of COVID-19 [14].

The causative agent of COVID-19, Severe Acute Respiratory Syndrome Coronavirus 2 (SARS-CoV-2), attaches its Spike protein to the angiotensin-converting enzyme 2 (ACE2) receptor of the host cell for internalization. This receptor is not only abundant in lung cells but, it is also present in astrocytes, oligodendrocytes, and neurons [15]. The virus accesses the CNS through retrograde axonal transport via olfactory bulb or the adjacent cribriform plate, vagus nerve, or through proximity by interacting with ACE2 receptors on the endothelial wall of brain capillaries and nearby neural cells [16]. The hematogenous and the "Trojan horse" routes are also described [15,17].

Although direct invasion of the CNS has been demonstrated for SARS-CoV-2 and also for its predecessors (SARS-CoV-1 and MERS-CoV) [17], an immune-mediated damage pathway is also described [18]. In this model, an exacerbated immune response and inflammatory process could lead to blood-brain barrier disruption, allowing transport of inflammatory cytokines such as IL-1β, TNF-α, and IL-6, and antibodies (like anti-myelin antibodies) to CNS, causing neurological damage even after the respiratory illness has subsided [2,3,19,20].

The presence of SARS-CoV-2 in CSF has been considered direct evidence of neuroinvasion [18,21,22]. However, most publications about neurological forms of COVID-19 are case reports in which viral tests in CSF were not performed or tested negative. Perhaps the lack of this information has led to greater acceptance of the immune-mediated mechanism for neurological COVID-19 [17,18,23]. But even in cases in which viral detection in CSF is absent, it should not rule out neuroinvasion given others imaging and CSF abnormalities have been described in these cases, as well as SARS-CoV-2 genome has also been reported in both human and animal model brain tissue [17,23].

There are others pathogenic mechanisms described for the association between COVID-19 and neurologic manifestations, such as reactivation of latent virus in CNS, angiotensin-renin system dysregulation with consequent thrombotic events leading to strokes [15,24], and the hypoxemic status resultant of respiratory distress leading to neural cells damage [14,25].

There are few prospective studies about neurological manifestations of COVID-19, especially those whose diagnosis was based on viral tests in CSF; besides, there are still many questions about CSF, radiographic patterns and pathways involved in the association between neurological disease andCOVID-19. Thus, we describe here clinical, imaging and CSF findings of a prospective cohort of patients with nasal positive tests for SARS-CoV-2 and neurological involvement, admitted in a tertiary center in Ribeirão Preto, Brazil, and discuss the pathogenic mechanisms related to these manifestations.

## Methods

### Casuistic

This study is part of an ongoing cohort research (beginning in 2018), which purpose is to evaluate the viral etiology of patients with acute neurological infections admitted at Ribeirão Preto Medical School Hospital, Brazil. It was approved by local and national ethical committees (registration number: 2.790.361), and formal consent was obtained from all patients or their legal guardians. The study was funded by Fundação de Amparo à Pesquisa de São Paulo (FAPESP).

Inclusion criteria were patients with acute neurological problems attributed to COVID-19 by health care providers, who were admitted between December 1st, 2020 and December 31, 2022. All of them tested positive for SARS-CoV-2 in the nasal swab, CSF lumbar puncture was performed in the first week of neurological symptoms, and another viral etiology was excluded following the ongoing cohort viral test protocol that included 29 distinct viruses. Patients who did not have confirmed COVID-19 and those who did not have cerebrospinal fluid collected in referred period were excluded. Nasal swabs were taken from all patients before admission, in accordance with local regulations at the time.

Clinical data was collected from medical records, including sex, age, comorbidities, neurological signs, clinical outcomes, syndromic diagnosis, occurrence of respiratory symptoms and its time interval from neurological manifestation. Complementary exams like imaging and cytological and biochemical CSF findings were also obtained. Due to the fact these patients are part of another ongoing study, they were followed until May 2023. None of the patients had a history of neurological disease or neurological symptoms.

### Viral test in CSF

All CSF samples were tested for qualitative RNA genome of SARS-CoV-2 by real-time transcriptase reverse-polymerase chain reaction (RT-PCR) utilizing the GeneFinder® COVID-19 FAST RealAmp kit (Osang HealthCare Co, Korea), approved for emergency use by Brazilian Health Agency (registration number 81914040006). The reaction was performed following the manufacturer's instructions. RNA was extracted from 200μl of CSF samples with Mini Spin Virus DNA/RNA Extraction Kit *Biopur extraction Kit* (Mobius LifeScience®) according to manufacturer's instructions.

### Analysis

The Chi-square test was used to compare percentages and means, respectively. Odds ratio with a 95% confidence interval was used to assess the association between clinical and laborator data, and the neurological syndrome presented by the patient, with the presence of

SARS-CoV-2 in CSF Significance was considered at p < 0.05. Descriptive multivariate analysis was performed.

## Results

A total of 27 patients were included in the study, all with suspected neurologic manifestations of COVID-19 and positive SARS-CoV-2 RNA in nasal swab specimens. Fourteen patients were male and 13 were female; age varied between 0-82 years (median 40.85), and most of them (n = 14) were between 36-50 years old, although there was no statistical significance in this finding.

Symptoms occurred concomitantly with positive nasal test or until 60 days after that (median 14 days). RT-PCR in spinal cord fluid (CSF-RT-PCR) resulted positive in 9 (33.3%) cases, five of them had a positive swab nasal test concomitant to neurologic disease. Among negative CSF-RT-PCR patients, six presented positive nasal tests during neurologic manifestations.

Clinical findings in the 27 patients included: paresis 11(40.7%), seizures 12 (44.4%), gait abnormalities 8 (29.6%), fever 8 (29.6%), headache 7 (25.9%), consciousness alteration 7 (25.9%), memory deficits 6 (22.2%), paresthesia 6 (22.2%), palsy 3 (11.1%), speech disorder 3 (11.1%), and less commonly visual impairment, neck stiffness and vomiting (one case of each (3.7%). The arithmetic means of symptoms presented was 3.8 per patient, (variation 1-7 symptoms) with no difference between positive or negative CSF- RT-PCR. CSF-RT-PCR was positive in 4 out 11 patients with paresis, 4 out 12 patients with epilepsy, 4 out 8 patients with gait abnormalities, 4 out 8 patients with fever, 1 out 7 patients with headache, 2 out 4 patients with paresthesia, 1 out 7 patients with consciousness alteration, 2 out 6 patients with memory or cognitive deficits and 2 out 3 with speech disorders. Those patients with a positive CSF-RT-PCR did not experience palsy, visual impairment, vomiting or delirium. Therefore, viral detection in CSF was not related to the neurological manifestations associated with COVID-19, as shown in Table 1. Respiratory complaints were expressed in 12 out 27 patients, five with positive CSF-RT-PCR. Comorbidities were significantly higher in the positive CSF-RT-PCR group (p < 0.006459), including hypertension, obesity and type 2 diabetes. Immunosuppressive condition was found in 2 cases: one positive and another negative CSF-RT-PCR.

Considering imaging findings, 14 patients had no abnormalities in nuclear magnetic resonance or tomography; eleven (40.7%) presented abnormalities like hypodensities, cerebral atrophy, foci of demyelination, inflammatory and hemorrhagic process; two cases had no imaging. Only 2 patients had imaging abnormalities that could be due to a previous undiagnosed condition, both with CSF-RT-PCR negative. None of them had previous imaging records nor neurological disease previously diagnosed. The neurological condition of all these patients was attributed to COVID-19 after excluding other possible causes. White cell counts in CSF were normal (reference: $\leq$ 5 cells/mm$^3$ after 1 month age) in all cases except for 3 adults: two of them were positive in CSF-RT-PCR (10 and 36 cells/mm$^3$) and one was negative (21 cells/mm$^3$). Glucose levels (reference: $\geq$ 50md/dl or $\leq$ 2/3 blood sugar result) were normal in all samples and protein levels were slightly elevated (reference: < 44 md/dl) in 12 samples: four patients with positive CSF-RT-PCR (69-85 mg/dl) and 8 with negative CSF-RT-PCR (49-469mg/dl). (Table 2).

Nineteen patients had poor clinical outcomes: two died and 17 evolved with sequelae or chronic conditions (5 positive and 12 negative CSF-RT-PCR testing). As shown in Table 3, syndromic diagnosis included stroke, epileptic manifestations, meningoencephalitis, encephalomyelitis, nonspecific dementia, Guillain-Barré Syndrome, optical neuritis, acute flaccid palsy, polyneuropathy, encephalitis and headache.

**Table 1. Clinical characterization of patients with neurologic manifestations of COVID- 19 according to CSF-RT-PCR results.**

| Characterization | Positive CSF-RT-PCR | Negative CSF-RT-PCR |
|---|---|---|
| Sex | | |
| Male | 6 (66.6%) | 8 (44.4%) |
| Female | 3 (33.3%) | 10 (66.6%) |
| Age | | |
| 0-18y | 2 (22.2%) | 4 (22.2%) |
| 19-35y | 2 (22.2%) | 2 (11.1%) |
| 36-60y | 4 (44.4%) | 10 (55.5%) |
| > 60y | 1 (11.1%) | 2 (11.1%) |
| Total | 9 (100%) | 18 (100%) |
| Clinical signs | | |
| Paresis | 4 (44.4%) | 7 (38.9%) |
| Seizures | 4 (44.4%) | 8 (44.4%) |
| Gait abnormalities | 4 (44.4%) | 4 (22.2%) |
| Fever | 4 (44.4%) | 4 (22.2%) |
| Headache | 1 (11.1%) | 6 (33.3%) |
| Paresthesia | 2 (22.2%) | 4 (22.2%) |
| Consciousness alteration | 1 (11.1%) | 6 (33.3%) |
| Memory or cognitive deficits | 2 (22.2%) | 4 (22.2%) |
| Palsy | none | 3 (16.6%) |
| Visual impairment | none | 1(5.5%) |
| Vomiting | none | 1(5.5%) |
| Speech disorder | 2 (22.2%) | 1(5.5%) |
| Neck stiffness | 1(11.1%) | none |
| Coma | 1 (11.1%) | 2 (11.1%) |
| Delirium | none | 3 (16.6%) |
| Respiratory symptoms | 5 (66.6%) | 7 (38.9%) |
| Immunosuppressive condition | 1 (11.1%) | 1(5.5%) |
| Comorbidities* | 8 (88.9%) | 6 (33.3%) |

*p < 0.0006459 (CI 95%).

## Discussion

In this prospective cohort study, all patients presenting distinct neurologic involvement associated with COVID-19 had their CSF samples tested for the SARS-CoV-2 genome. It drew our attention that most publications on neurologic manifestations associated with COVID-19 are case reports or review articles in which viral tests in CSF are not described. Our study brings together CSF-RT-PCR, clinical, imaging and CSF features. Twenty-seven patients participated in this study, one of the most significant samples reported [22,26,27].

Detection rates of SARS-CoV-2 in CSF samples are low, ranging from 4.76 to 9% of samples tested [21,27,28]. Our positivity rate to SARS-CoV-2 in CSF reached 33.3%, much higher than has been reported. This result is probably related to the viral load in CSF, the time interval between the appearance of the first symptoms and lumbar puncture, or the high sensitivity of the PCR kit we utilized (limit of detection of 2 copies/µl for gene N and five copies/µl for genes RdRP and E). The lack of standardized protocols may also be the cause of discrepant results [15,25].

**Table 2. Imaging and CSF findings in patients with neurologic manifestation of COVID-19, according to SARS-CoV-2 CSF-RT-PCR result.**

| Findings | Positive CSF PCR | Negative CSF PCR |
|---|---|---|
| Imaging | | |
| Normal | 2 (22.2%) | 12(66.6%) |
| Alterated | 6 (66.6%) | 5 (27.8%) |
| No record | 1(11.1%) | 1(5.6%) |
| CSF white cell count | | |
| 0-5cel/mm3 | 6 (66.6%) | 17 (94.4%) |
| 6-10 cel/mm3 | 1(11.1%) | none |
| 11-50 cel/mm3 | 1(11.1%) | 1(5.6%) |
| Not counted | 1(11.1%) | none |
| CSF protein | | |
| Normal | 5 (55.6%) | 10 (66.6%) |
| Elevated | 4 (44.4%) | 8 (44.4%) |
| Total | 9 (100%) | 18 (100%) |

**Table 3. Syndromic diagnosis of patients with neurological forms of COVID-19 according to CSF-RT-PCR result.**

| Diagnosis | Positive CSF-RT-PCR | Negative CSF-RT-PCR |
|---|---|---|
| | Positive | Negative |
| Stroke | 2 (22.2%) | 1 (5.5%) |
| Nonspecific dementia | 2 (22.2%) | 2(11.1%) |
| Seizures/epilepsy | 2 (22.2%) | 6 (33.3%) |
| Guillain-Barré Syndrome | 1(11.1%) | 1 (5.5%) |
| Meningoencephalitis | 1(11.1%) | none |
| Encephalomyelitis | 1(11.1%) | none |
| Acute flaccid palsy | none | 2 (11.1%) |
| Polyneuropathy | none | 1 (5.5%) |
| Optical neuritis | none | 1 (5.5%) |
| Headache | none | 1 (5.5%) |
| Encephalitis | none | 2(11.1%) |
| Meningitic signs | none | 1 (5.5%) |
| Total | 9 (100%) | 18 (100%) |

There is a wide range of neurologic manifestations associated with COVID-19, occurring from isolated signs to well-known neurological syndromes that evolve from distinct pathogenic mechanisms [4,8,10,11,29,30]. We could not correlate such manifestations with the time interval between respiratory symptoms (when they existed) or viral detection in nasal swabs, as similar manifestations could be observed either concomitant with COVID-19 or after convalescence of the disease. Indeed, respiratory symptoms do not seem to be essential to neurological involvement, since some patients did not have any respiratory complaints, and the nasal swab was the unique evidence of COVID-19 [31]. Thus, it is probable that hypoxia, as a result of respiratory failure, would damage neural cells and lead to neurological disease, which is not a pathogenic pathway applicable to all cases [14,32].

Neurologic manifestations are described in patients with prolonged disease or late-onset sequelae of COVID-19 and they are called "Long Covid" [31]. These manifestations include brainstorm, fatigue, anosmia, dysgeusia, cognitive or memory loss, psychiatric disorders, motor autonomic or sensitive impairment, sleep disorders, Guillain- Barré Syndrome; it occurs in about one-third of COVID-19 survivors [33]. The same mechanisms involved in neurologic complications of COVID-19 are engaged in Long Covid [25,31,33]. Although neural direct invasion has been considered in the pathogenesis of Long Covid, viral persistence in CSF has not been documented [25,33]. Also, our study does not have sufficient data to confirm or deny this hypothesis.

Some authors describe that the pathogenesis of neurological involvement of COVID-19 is related to the time interval between the disease and the moment which the neurological symptoms arise: if neurological and respiratory symptoms are simultaneous, the neural invasion of SARS-CoV-2 is the mechanism more plausible, but if neurological signs arise latterly, the immune-mediated mechanism could be the causative [27,28,34]. Our data does not support this theory about the pathogenesis of COVID-19 affecting CNS, because the viral genome could be detected in CSF samples either concomitant or after COVID-19.

Detection of SARS-CoV-2 in CSF has been firmly associated with evidence of neuroinvasion [13,22]. As the detection rates in CSF described elsewhere are low, some researchers have discouraged routine tests and considered that the pathogenesis of COVID-19 and neurologic involvement could be immune-mediated. Our 33% detection rate of SARS-CoV-2 in CSF samples suggests that these tests are valid and more studies are necessary to better understand the timing and viral load of SAR-CoV-2 in CSF.

Many pathogenic ways are described associating COVID-19 and neurological disease (neuroinvasion, cytokine storm, immune-mediated mechanism, thrombotic events, cross-reaction or production of autoantibodies, effects of hypoxia status) and probably one or more of these mechanisms might be present in each case [16,25,35–37]. Until now, none of these explanations could be applied to all situations. The pathogenic pathway might be more related to the final syndromic diagnosis presented for each patient, like the production of autoantibodies observed in cases of demyelinating syndromes with anti-myelin antibodies detection, or the thrombotic events in patients with stroke or central venous thrombosis in brain.

Interestingly, it was not possible to correlate any imaging alteration to neurologic manifestations in COVID-19, neither biochemical nor cytologic pattern in CSF [24,38–40]. Although more neurological findings were observed in positive CSF-RT-PCR cases, it was not significant.

This study has some limitations, like the small sample size, localization in one single center, and the presence of a low number of immunosuppressed patients. Besides, inflammatory/ immune markers, like oligoclonal bands, immunoglobulin G index in CSF, auto-antibodies like anti-myelin and aquaporin 4, and interleukin-6 were not performed in all patients. The neurological condition of all these patients was attributed to COVID-19. Some patients presented neurological signs a long time after testing positive for SARS-CoV-2 (up to 60 days). Although it may be questioned whether COVID-19 was really the cause of the neurological condition, other potential causes have been ruled out previously.

However, it is a cohort study on neurologic manifestations of COVID-19 that includes the detection of SARS-CoV-2 in CSF associated with clinical, imaging, and CSF parameters. Our results suggest that SARS-CoV-2 indeed invades CNS and more studies are necessary to better understand this mechanism.

## Conclusion

In conclusion, the patients in the present study had COVID-19 and neurologic involvement including a wide range of clinical manifestations. SARS-CoV-2 was detected in one-third of CSF samples, regardless of the time interval between COVID-19 and the onset of neurological signs. The neuroinvasion potential of SARS-CoV-2 should be more studied.

## Acknowledgments

We express our sincere gratitude to all patients who agreed to participate in this study and the medical staff who performed lumbar punctures and saved some CSF. We sincerely thank Dr. Tissiana Marques de Haes who made all the samples available and stored them until they were collected for the research.

## Author contributions

**Conceptualization:** Carla de Oliveira Cardoso.

**Data curation:** Carla de Oliveira Cardoso.

**Formal analysis:** Carla de Oliveira Cardoso, Evandra Strazza Rodrigues Sandoval.

**Investigation:** Carla de Oliveira Cardoso, Evandra Strazza Rodrigues Sandoval, Lilian Beatriz Moreira de Oliveira Chagas, Soraya Jabur Badra.

**Methodology:** Carla de Oliveira Cardoso.

**Project administration:** Luiz Tadeu Moraes Figueiredo.

**Resources:** Dimas Tadeu Covas, Simone Kashima Haddad.

**Supervision:** Evandra Strazza Rodrigues Sandoval, Simone Kashima Haddad, Luiz Tadeu Moraes Figueiredo.

**Validation:** Carla de Oliveira Cardoso.

**Visualization:** Carla de Oliveira Cardoso, Evandra Strazza Rodrigues Sandoval.

**Writing – original draft:** Carla de Oliveira Cardoso.

**Writing – review & editing:** Carla de Oliveira Cardoso, Evandra Strazza Rodrigues Sandoval, Lilian Beatriz Moreira de Oliveira Chagas, Soraya Jabur Badra, Dimas Tadeu Covas, Simone Kashima Haddad, Luiz Tadeu Moraes Figueiredo.

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
