## [Decision Letter · Decision Letter 0]

2 Jan 2024

PONE-D-23-32540Neurologic manifestations of COVID-19 and viral test in cerebrospinal fluid.PLOS ONE

Dear Dr. Cardoso,

Thank you for submitting your manuscript to PLOS ONE. After careful consideration, we feel that it has merit but does not fully meet PLOS ONE’s publication criteria as it currently stands. Therefore, we invite you to submit a revised version of the manuscript that addresses the points raised during the review process.

We look forward to receiving your revised manuscript.

Kind regards,

Kartikeya Rajdev, MD

Academic Editor

PLOS ONE

“no. The funders had no role in study design, data collection and analysis, decision to publish, or preparation of the manuscript.”

Reviewers' comments:

Reviewer's Responses to Questions

**Comments to the Author**

1. Is the manuscript technically sound, and do the data support the conclusions?

Reviewer #1: Yes

Reviewer #2: Yes

Reviewer #3: Yes

2. Has the statistical analysis been performed appropriately and rigorously? 

Reviewer #1: Yes

Reviewer #2: Yes

Reviewer #3: Yes

3. Have the authors made all data underlying the findings in their manuscript fully available?

Reviewer #1: Yes

Reviewer #2: Yes

Reviewer #3: Yes

4. Is the manuscript presented in an intelligible fashion and written in standard English?

Reviewer #1: Yes

Reviewer #2: No

Reviewer #3: Yes

5. Review Comments to the Author

Reviewer #1: The authors have made a great attempt at looking at the neurologic manifestations of COVID-19 and correlating with CSF findings as well as respiratory symptoms. Barring a few minor clarifications listed below, overall, it is a good research study.

1. What are the exclusion criteria of the study? I am curious to know especially since the study could only include 27 patients with neurologic manifestations of COVID-19 over 2 years. Likely there were many more that couldn't get an LP done or anything else.

2. While 27 patients were included in the study, only 12 had respiratory symptoms. What was the indication of nasal swabbing the other 15? Was it purely based on neurologic symptoms or was it routinely being done on all admissions during the pandemic?

3. While mentioning elevated/normal levels of glucose and protein, it would be better to mention what range was considered normal and what was considered elevated.

4. While the table mentioned ages starting at 0, the description in line 155 mentioned ages starting at 1. Kindly rectify. Also, since CSF WBC counts would differ in infants, what are the ages of the 3 patients with elevated WBC counts? If it is an age 0 patient, is it really an elevated count?

5. Minor languages at various places including but not limited to the sentences starting in lines 34, 39, 72, 102, 107, 112, 115, 216, 219.

I appreciate the efforts of the researchers.

Reviewer #2: Neurological manifestation of COVID-19 reviews:

I congratulate authors for performing this study. Patient evaluations in extensive, neuroimaging and CSF RT-PCR was performed in all cases. I have carefully reviewed this manuscript and I will suggest following edits.

Abstract: Line 33- “as well there are still many questions about COVID-19/neurological disease” I will suggest removing / used between COVID 19 and neurological disease.

Line 39- “presentation, interval between COVID-19 e onset of neurological alterations, syndromic” what is the meaning of e after COVID-19? Please make corrections.

Line 45, 46, 47- “two patients had elevated counts and viral detection in CSF (10 and 36 cells/mm3) and one also had elevated count but viral detection in CSF was negative (21cells/mm3).” When mentioning elevated CSF counts, please clearly write elevated CSF WBC counts.

Introduction:

Lines

74 Neurological manifestations are present in about one-third of

75 COVID-19 cases, ranging from mild symptoms, such as anosmia, to more severe forms

76 like encephalitis and demyelinating syndromes [6].

Reference 6 cited by authors is a study evaluating neurological manifestations of SARS CoV-2. This study did not notice 1/3rd of the COVID-19 patients having neurological symptoms. In cited study out of the 1263 studied patients 43 were noted to have encephalopathy. The refence article [6] does provide reference to a study published in JAMA, 1/3rd of COVID-19 patients in this study noted to have neurological symptoms. Please see following reference

Mao L, Jin H, Wang M, Hu Y, Chen S, He Q, Chang J, Hong C, Zhou Y, Wang D, Miao X, Li Y, Hu B. Neurologic Manifestations of Hospitalized Patients With Coronavirus Disease 2019 in Wuhan, China. JAMA Neurol. 2020 Jun 1;77(6):683-690. doi: 10.1001/jamaneurol.2020.1127. PMID: 32275288; PMCID: PMC7149362.

Lines

114 many questions about CSF, radiographic patterns and pathways involved in the COVID-

115 19/neurological disease association.

Again, I will suggest changing COVID-19/Neurological disease to COVID-19 related neurological disease.

Methods: Authors included all admitted patients with acute neurological problems attributed to COVID-19 between Dec 1 2020 to Dec 31st 2022. Lumbar puncture was performed in all patients included in this study. The study was approved by the ethics committee and consent was obtained from all patients. Clinical data was collected from medical records, including sex, age, comorbidities, neurological signs, clinical outcomes, syndromic diagnosis, occurrence of respiratory, symptoms and its time interval from neurological manifestation. Complementary exams like imaging and cytological and biochemical CSF findings were also obtained.

In the methods section please mention how long after initial diagnosis of COVID-19 these patients were followed?

Funding?

Analysis: Please mention if univariate or multivariate analysis was performed. As you have mentioned associations in your result section.

Results:

Lines

163 Clinical findings in the 27 patients included paresis (40.7%), seizures (44.4%), gait

164 abnormalities (29.6%), fever (29.6%), headache (25.9%), paresthesia (25.9%),

165 consciousness alteration (25.9%), memory deficits (22.2%), palsy (11.1%), speech

166 disorder (11.1%),

Please mention actual numbers in front of percentages for individual percentages. Also few of the patients may have multiple neurological symptoms please describe those patients.

Lines

167(3.7%). Viral detection in CSF was not related to the neurological manifestations

168 associated with COVID-19, as shown in table 1.

Please describe table findings in details here such as out of 11 patients with COVID-19 related paresis only 4 had a positive CSF RT-PCR.

Lines

169 in 12/27 patients change to 12 out of 27 patients.

170 higher in the positive CSF-RT-PCR group (p=0,006459). Please correct P value number here

176 Considering imaging findings, 14 patients had no abnormalities in nuclear magnetic

177 resonance or tomography; eleven (40.7%) presented abnormalities like hypodensities,

178 cerebral atrophy, foci of demyelination, inflammatory and hemorrhagic process; two

179 cases had no imaging.

Please also mention these neuro imaging findings could also be co-incidental and may not be related to direct effect of COVID-19. Considering

Discussion:

Lines

202 study brings together CSF-RT-PCR, clinical, imaging e CSF features. What does e denote here?

211 There is a wide range of neurologic manifestations associated with COVID-19, occurring

212 since isolated signs to well-known neurological syndromes which evolve from distinct

213 pathogenic mechanisms.

In line 212 change since to from.

Lines

216Indeed, respiratory alterations does

217 not seem to be essential to neurological involvement, once some patients did not have any

218 respiratory complaints, and the nasal swab was the unique evidence of COVID-19.

Please make corrections in this sentence. Consider changing respiratory alterations to respiratory symptoms. Line 217 “once some patients” is not appropriate, please make correction here.

219 Thus, is probable that hypoxia, as resultant of respiratory distress, would damage neural

220 cells and lead to neurological disease, is not a pathogenic pathway applicable to all cases

221 [14,31]

Please make corrections Thus, it is probable.. also consider changing respiratory distress to respiratory failure?

226 latterly, the immuno-mediated mechanism could be the causative [26,27,32].

Please change immuno-mediated to immune-mediated

234 rate of SARS-CoV-2 in CSF samples suggests that these tests are valid and maybe more

235 studies are necessary to better comprehend the timing and viral load of SAR-CoV-2 in

236 CSF.

In line 234 please change maybe more studies are necessary to (remove may be) more studies are necessary to better understand (remove comprehend)

Strength: Patient evaluations in extensive, neuroimaging and CSF RT-PCR was performed in all cases.

Weakness: Small sample size, single center. Authors have acknowledged limitations of the study in the discussion section.

I would suggest authors briefly discuss long COVID in the discussion section of this article. There is growing data suggesting lingering neurological symptoms including brain fog, anxiety, depression, suicidal ideation, chronic fatigue due to long term effects of COVID-19. Pathological mechanism of these symptoms is very poorly understood.

This article also needs significant grammatical corrections before publishing.

Reviewer #3: This is a very good study trying to evaluate the association of COVID-19 with Neurological syndromes and assess the pathogenic mechanisms.

It is interesting to see that COVID RT-PCR was positive in about one third of patients and though significance of it in causation of neurological symptoms is unclear but I agree neuroinvasion potential of SARS-CoV-2 needs to be studied more.

The study is limited by its size and being confined to one centre in Brazil and multi site study with bigger sample size is needed.

It would have been helpful if authors tried to see if the patients selected for this study had any preexisting neurological conditions such as stroke, neuropathy or demyelinating diseases as very often symptoms could be due to recrudescence in the setting of systemic infection, hypoxia or metabolic derangement which can happen in the setting of COVID-19. I agree that studies would need to look other demyelinating diseases markers along with SARS-CoV-2 as it could have possibly helped with differentiation between neuroinvasion or inflammatory mediated mechanisms.

I understand the study was done between December 2020 and December 2021 and COVID vaccine was not available by then and future studies should also look into vaccination status in patient selection.

Over all, Its a great study and I recommend with out any revisions.

6. PLOS authors have the option to publish the peer review history of their article (what does this mean? ). If published, this will include your full peer review and any attached files.

**Do you want your identity to be public for this peer review?** For information about this choice, including consent withdrawal, please see our Privacy Policy .

Reviewer #1: No

Reviewer #2: **Yes: ** Rajendra Karnatak

Reviewer #3: **Yes: ** Pradeep K R Kumbham

---

## [Author Response · Author response to Decision Letter 1]

16 Feb 2024

Ribeirão Preto, February 15, 2024

Staff Editors

Plos One

Dear Editors:

We have submitted our manuscript Neurologic manifestations of COVID-19 and viral test in cerebrospinal fluid, where we studied the neurological involvement in COVID-19 and the presence of SARS-CoV-2 in cerebrospinal fluid, besides its association with clinical features, syndromic diagnosis, radiographic findings, biochemical and cell parameters in CSF.

Here, we response to Journal requirements:

1- We have adjusted file names to meet Plos One’s style requirement;

2- A) The cohort study was supported by Fundação de Amparo à Pesquisa de São Paulo, which maintains almost all researches from Virology Research Center. The Foundation is not involved in the design of the project, the collection or analysis of data, and is not involved in the publication of the final work. Neither I nor the other authors have been paid by this foundation for their work;

B) “the funders had no role in study design, data collection and analysis, decision to publish, or preparation of the manuscript”;

C, D) “the authors received no specific funding for this work”;

3- We confirm that all data are within the manuscript, and also confirm the reviewers;

4- The reference list was updated.

Here, we answer the Reviewer’s considerations:

Reviewer 1

1- The exclusion criteria were patients who have not been diagnosed with COVID-19 and patients who refused to participate in the study. Patients were included in the study group at the time we received CSF samples, so patients whose CSF was not collected were excluded;

2- During the pandemia, the nasal test was carried out for all patients admitted to the hospital according to internal procedures. This information was included in the text;

3- The reference parameters for glucose, protein and cell in the CSF were included in the text;

4- The table and line 155 have been modified. All three patients with CSF leukocytosis were adults and these data are included in the text as a reference value;

5- The errors have been identified and we hope that they will all be corrected.

Reviewer 2

1- Abstract: lines 33, 39, 45, 46 and 47 were fixed.

2- Introduction

- Lines 74, 75, 76 and 115 were fixed;

- Reference 6 has been fixed, sorry for the mistake.

3- Method

- Given the cohort criteria, patients from this work were followed up until May 2023. This information is presented in the section;

- Funding: information was included in the text; this work was funded by the Fundação de Amparo à Pesquisa de São Paulo.

4- Analysis: Descriptive multivariate analysis was performed and information was added in the text.

5- Results

- Fixed lines 163, 164, 165, 166, 167, 168, 169, 170, 176, 177, 178 and 179;

- Patients showed an average of 3.8 symptoms with no difference between groups. Further details are given in the text;

- Two imaging abnormalities may be the result of previously undiagnosed injury, both patients were negative in CSF-RT-PCR. The information is described in the text.

6- Discussion

- Lines 202, 211, 212, 213, 216, 217, 218, 219, 220, 221, 226 and 234 were modified;

- Gramatical mistakes were fixed.

Reviewer 3

- None patient had previous neurologic condition diagnosed, and none of the neurological manifestations associated with COVID-19 were considered a recrudescence by the assistant staff. This information was added in the text.

We sincerely thank all the reviewer’s contributions and we expected all the mistakes of this work were solved.

I hope that my clarifications have resolved any doubts that may exist and I am at your disposal for further information.

Please let us know if you need further information.

Yours sincerely,

Carla de Oliveira Cardoso

Virology Research Center,

Department of Internal Medicine

Ribeirão Preto Medical School

University of São Paulo,

Ribeirão Preto, São Paulo, Brazil.

Phone: 55 34 99103 9775

carla_cardoso@hotmail.com

---

## [Decision Letter · Decision Letter 1]

30 Apr 2024

PONE-D-23-32540R1Neurologic manifestations of COVID-19 and viral test in cerebrospinal fluid.PLOS ONE

Dear Dr. Cardoso,

Thank you for submitting your manuscript to PLOS ONE. After careful consideration, we feel that it has merit but does not fully meet PLOS ONE’s publication criteria as it currently stands. Therefore, we invite you to submit a revised version of the manuscript that addresses the points raised during the review process.

We look forward to receiving your revised manuscript.

Kind regards,

Kartikeya Rajdev, MD

Academic Editor

PLOS ONE

Journal Requirements:

Reviewers' comments:

Reviewer's Responses to Questions

**Comments to the Author**

1. If the authors have adequately addressed your comments raised in a previous round of review and you feel that this manuscript is now acceptable for publication, you may indicate that here to bypass the “Comments to the Author” section, enter your conflict of interest statement in the “Confidential to Editor” section, and submit your "Accept" recommendation.

Reviewer #1: All comments have been addressed

Reviewer #2: All comments have been addressed

Reviewer #3: All comments have been addressed

2. Is the manuscript technically sound, and do the data support the conclusions?

Reviewer #1: Yes

Reviewer #2: Yes

Reviewer #3: Yes

3. Has the statistical analysis been performed appropriately and rigorously? 

Reviewer #1: Yes

Reviewer #2: Yes

Reviewer #3: Yes

4. Have the authors made all data underlying the findings in their manuscript fully available?

Reviewer #1: Yes

Reviewer #2: Yes

Reviewer #3: Yes

5. Is the manuscript presented in an intelligible fashion and written in standard English?

Reviewer #1: Yes

Reviewer #2: Yes

Reviewer #3: Yes

6. Review Comments to the Author

Reviewer #1: (No Response)

Reviewer #2: Line 185 (p<0,006459), including hypertension, obesity and type 2 diabetes. Immunosuppressive. Please make corrections here in p value.

201 Table 2- Imaging and CSF findings in patients with neurologic manifestation of COVID-

202 19, according to SARS-CoV-2 CSF-RT-PCR result. Please also mention these neuro imaging findings could also be co-incidental and may not be related to direct effect of COVID-19. If there were no CNS imaging to compare if these findings were new.

Strength: Patient evaluations in extensive, neuroimaging and CSF RT-PCR was performed in all cases.

Weakness: Small sample size, single center. Authors have acknowledged limitations of the study in the discussion section.

I would suggest authors briefly discuss long COVID in the discussion section of this article. There is growing data suggesting lingering neurological symptoms including brain fog, anxiety, depression, suicidal ideation, chronic fatigue due to long term effects of COVID-19. Pathological mechanism of these symptoms is very poorly understood.

Reviewer #3: Thank you for giving me the opportunity to revie the case report and it is a great case report. I recommend for publication.

7. PLOS authors have the option to publish the peer review history of their article (what does this mean? ). If published, this will include your full peer review and any attached files.

**Do you want your identity to be public for this peer review?** For information about this choice, including consent withdrawal, please see our Privacy Policy .

Reviewer #1: No

Reviewer #2: **Yes: ** Rajendra Karnatak

Reviewer #3: **Yes: ** Pradeep Kumbham

---

## [Author Response · Author response to Decision Letter 2]

14 Jun 2024

Reviewer 2: line 185 was fixed.

Information about image findings, previous neurological disease or previous imaging findings were explained in text under Table2.

A paragraph briefly discussing long covid was added in discussion section.

We sincerely thank your generous contributions for this paper.

Carla Cardoso, phd.

---

## [Decision Letter · Decision Letter 2]

29 Jul 2024

PONE-D-23-32540R2Neurologic manifestations of COVID-19 and viral test in cerebrospinal fluid.PLOS ONE

Dear Dr. Oliveira Cardoso,

Thank you for submitting your manuscript to PLOS ONE. After careful consideration, we feel that it has merit but does not fully meet PLOS ONE’s publication criteria as it currently stands. Therefore, we invite you to submit a revised version of the manuscript that addresses the points raised during the review process.

We look forward to receiving your revised manuscript.

Kind regards,

Kartikeya Rajdev, MD

Academic Editor

PLOS ONE

Journal Requirements:

Reviewers' comments:

Reviewer's Responses to Questions

**Comments to the Author**

1. If the authors have adequately addressed your comments raised in a previous round of review and you feel that this manuscript is now acceptable for publication, you may indicate that here to bypass the “Comments to the Author” section, enter your conflict of interest statement in the “Confidential to Editor” section, and submit your "Accept" recommendation.

Reviewer #2: All comments have been addressed

2. Is the manuscript technically sound, and do the data support the conclusions?

Reviewer #2: Yes

3. Has the statistical analysis been performed appropriately and rigorously? 

Reviewer #2: Yes

4. Have the authors made all data underlying the findings in their manuscript fully available?

Reviewer #2: Yes

5. Is the manuscript presented in an intelligible fashion and written in standard English?

Reviewer #2: Yes

6. Review Comments to the Author

Reviewer #2: I congratulate authors to make appropriate corrections in manuscript and submitting an updated version of the article. In the updated version I suggest to make following corrections:

Methods:

133 “Inclusion criteria were patients with acute neurological problems attributed to COVID-

134 19 by health care assistants, who were admitted between December 1st, 2020 and

135 December 31, 2022.” Please correct this sentence what is a health care assistant? Do you mean healthcare provider?

Still not clear how authors concluded a neurological symptom that happened 60 days after the COVID-19 nasal swab PCR was attributed to COVID-19 infection related. Too long of a period to confidently attribute COVID-19 for neurological symptoms. Authors should address this in the weakness section of this article. It cannot be confidently said neurological symptoms this far out is due to COVID-19 infection but there was no alternative etiology of neurological symptoms was found in these patients.

7. PLOS authors have the option to publish the peer review history of their article (what does this mean? ). If published, this will include your full peer review and any attached files.

**Do you want your identity to be public for this peer review?** For information about this choice, including consent withdrawal, please see our Privacy Policy .

Reviewer #2: **Yes: ** RAJENDRA KARNATAK

---

## [Author Response · Author response to Decision Letter 3]

12 Sep 2024

Lines 133-135 were fixed. Limitations of the study were verified.

---

## [Editor Report · Decision Letter 3]

10 Oct 2024

Neurologic manifestations of COVID-19 and viral test in cerebrospinal fluid.

PONE-D-23-32540R3

Dear Dr. Cardoso,

We’re pleased to inform you that your manuscript has been judged scientifically suitable for publication and will be formally accepted for publication once it meets all outstanding technical requirements.

Kind regards,

Kartikeya Rajdev, MD

Academic Editor

PLOS ONE

Additional Editor Comments (optional):

The authors have addressed the comments from the reviewers. 
---

## [Editor Report · Acceptance letter]

PONE-D-23-32540R3

PLOS ONE

Dear Dr. Cardoso,

I'm pleased to inform you that your manuscript has been deemed suitable for publication in PLOS ONE. Congratulations! Your manuscript is now being handed over to our production team.

Kind regards,

on behalf of

Dr. Kartikeya Rajdev

Academic Editor

PLOS ONE